# Self-Control of Inflammation during Tail Regeneration of Lizards

**DOI:** 10.3390/jdb9040048

**Published:** 2021-11-02

**Authors:** Bingqiang He, Honghua Song, Yongjun Wang

**Affiliations:** 1Key Laboratory of Neuroregeneration of Jiangsu and Ministry of Education, Co-Innovation Center of Neuroregeneration, Nantong University, 19 Qixiu Road, Nantong 226001, China; hebingqiang107@163.com (B.H.); song_hong_hua@126.com (H.S.); 2School of Medicine, Nantong University, 19 Qixiu Road, Nantong 226001, China

**Keywords:** lizard, regeneration, appendage, blastemal cells, inflammation

## Abstract

Lizards can spontaneously regenerate their lost tail without evoking excessive inflammation at the damaged site. In contrast, tissue/organ injury of its mammalian counterparts results in wound healing with a formation of a fibrotic scar due to uncontrolled activation of inflammatory responses. Unveiling the mechanism of self-limited inflammation occurring in the regeneration of a lizard tail will provide clues for a therapeutic alternative to tissue injury. The present review provides an overview of aspects of rapid wound healing and roles of antibacterial peptides, effects of leukocytes on the tail regeneration, self-blocking of the inflammatory activation in leukocytes, as well as inflammatory resistance of blastemal cells or immature somatic cells during lizard tail regeneration. These mechanistic insights of self-control of inflammation during lizard tail regeneration may lead in the future to the development of therapeutic strategies to fight injury-induced inflammation.

## 1. Introduction

Several adult vertebrates can spontaneously regenerate their lost appendages in high fidelity, while such regenerative capacity is evolutionarily absent in mammals. Instead, appendage amputation of adult mammals often heals the wounding by the formation of a fibrotic scar [1,2,3,4,5]. The observation of the appendage regenerating process in vertebrates reveals that epimorphic regeneration shares conserved histological events, including re-epithelization of the wound, formation of the blastema and growth of the different tissues [6,7,8,9]. 

The appearance of transient myofibroblasts and the formation of a highly proliferative wound epithelium is the main trait in early wound healing. The wound is rapidly sealed without the formation of scar tissue. After two days post-amputation (DPA), a wound epithelium begins to grow and re-epithelialisation is complete within 5 DPA [7,10]. After 7–10 DPA, the blastema derived from dedifferentiated tissue or existing progenitor cells emerges near the distal end of the rupture site and gradually differentiates to form a replacement appendage after 30 DPA [10,11,12]. Accordingly, the underlying cellular and molecular mechanisms that govern the reconstruction of multi-tissue structures are highly overlapped [8,13,14]. 

To date, many researches focus on the anamniotes, including teleosts and urodeles, in an attempt to decipher the key regulators that influence the regenerative process of fins, limbs and tails, while the amniotic reptiles at important phylogenetic position receive little attention with respect to their robust ability in tail regeneration. Lizards have evolved a successful strategy to escape from predation by voluntarily shedding or autotomizing their tail, and generally, it takes months or years to regenerate a replica containing nerves, blood vessels, skeletal muscles and cartilage [6,7,15]. To avoid possible infection at diverse ecology and/or the endogenous activation of inflammation that will result in the failure of the regeneration, the lizards are obliged to employ an efficient molecular or genetic mechanism to rapidly control excessive activation of inflammatory response following tail amputation. In this review, we will introduce several aspects that lizards have evolved to circumvent inflammation during the spontaneous tail regeneration, which may provide a promising therapeutic alternative to mammalian traumatic inflammation.

## 2. Rapid Wound Healing and Protection of Antibacterial Peptides

Scar-free wound healing in lizards is rapidly initiated after tail amputation, which stabilizes wound and protects against infection or desiccation. Such damage of the tail causes minimal bleeding, attributing to the special tissue structure at the sixth caudal vertebra (-*Eublepharis macularius*), where the vertebra can be split and the arterial sphincter can rapidly contract. [15]. Within 0.5 DPA, the wound site develops a clot of blood, tissue fluid and tissue debris [7]. By 7–14 DPA, the clot dissolves and reveals a smooth wound epithelium, under which the blastema begins to differentiate to new tissues [7,15,16,17]. Notably, the natural hemostasis of the wounded tail is achieved within seconds, much faster than those of mammals such as rats. We have found that the higher enzymatic activity of lizard thrombin is responsible for procoagulation (manuscript in submission). The rapid re-epithelialization of the wounds significantly reduces the risks of microorganism invasion and subsequent infection or inflammation, providing a favorable regenerative microenvironment [17].

Antimicrobial peptides (AMPs) are an important protective barrier for lizards in terms of anti-infection following tail amputation, which help these animals survive from challenging environments filled with bacteria [18,19]. Up until now, two types of antimicrobial peptides have been isolated and characterized from lizards, including defensin and chatelicidin [20]. The well-known defensins contain three subclasses, including α, β- and θ-defensins [21,22], but only β-defensins have been identified in lizards that are defined as having a bonding pattern of cysteines (Cys1–Cys5, Cys2–Cys4, and Cys3–Cys6) [22]. Genomic analysis demonstrates that a total of 32 genes encoding different β-defensin-like peptides are deposited in the genome of the lizard *Anolis carolinensis* and likely also in other lizards [22]. Tail amputation induces the expression of β-defensins within the granulocytes around the wound, where they are distributed in the phagosomes containing the degraded bacteria [19]. As a consequence, the high production of these β-defensins, such as *A. carolinensis* beta defensins (AcBD15) and AcBD27 is correlated with the absence of inflammation for the wounded tail [23,24], indicating the physiological importance of the β-defensins in protecting the exposed tissues from microbial invasion and occurrence of inflammation.

Cathelicidins are the other crucial antimicrobial peptides identified in lizards. Originally, they were stored in the azurophilic granules of neutrophils as the inactive prepropeptide before undergoing proteolytic cleavage by neutrophil elastase or serine protease [25,26,27]. The mature or active peptides are 12–80 amino acids long and encoded by exon 4 of the cathelicidin gene [28]. A total of four cathelicidin members, including cathelicidins 1 (CATH-1), -2a, -2b, and -3, are characterized in the genome of the *Carolina anole* lizard [29], but the mature forms of CATH-2a and -2b are the same [30]. CATH-1 and -2 are normally stored in granulocytes in the blood or in connective tissues of lizards, while keratinocytes can be induced to produce and possibly release these molecules in response to injury or microbial invasion [30,31]. As for CATH-3, there is still less information about its function, except a study showing that it also contributes to killing gram negative and gram positive bacteria at high dosage [32]. Taken together, both defensins and cathelicidins are key natural barriers of anti-bacteria and anti-inflammation following lizard tail amputation [33].

## 3. Infiltration of Leukocytes to the Wounded Tail

The lower inflammatory response of the wounded tail does not mean a lack of leukocyte infiltration at the injured sites of lizards. By contrast, wound healing is associated with leukocytic response without activation of excessive inflammation [6]. Injury to the tail leads to the disruption of vascular network system, which immediately triggers aggregation and activation of platelets to begin the clotting cascade. During this process, chemokines and cytokines produced by local cells and platelet degranulation recruit peripheral leukocytes to wound sites [34,35,36,37]. Generally, neutrophils infiltration occurs within several minutes, sequentially followed by monocytes and lymphocytes [38]. Besides functioning in phagocytosis and killing bacteria, they also release proteases, chemokines, and growth factors which will stimulate cell proliferation and migration within the damaged tissues [39,40]. After the loss of the tail, heterophils, a kind of granulocytes equivalent to the mammalian neutrophils, are recruited to the wound site to participate in phagocytosis of tissue debris and defense of various microbes [23,41,42,43]. Immunostaining reveals that these heterophils upregulate the expression of β-defensins, including AcBD15, to inhibit rather than trigger the inflammation [23].

The tail amputation of lizards inevitably triggers monocytes/macrophages to participate in the process of wound healing and subsequent regeneration. A large amount of macrophages have been observed within the damaged tissues or underneath the wounded epithelium following tail loss [23,43]. Though few mechanisms of the injury-induced recruitment of macrophages have been elucidated, the proinflammatory cytokines or chemokines are undoubtedly the main contributors of such cell events [34,35,36,37,43]. For example, macrophage migration inhibitory factor (MIF) derived from spinal cord plays a key role in the recruitment of the macrophages to the injured sites of spinal cord following gecko tail amputation. The recruited macrophages have been found to be involved in the phagocytosis of pathogens or tissue debris. Experiments on both lizards and amphibian models have clearly demonstrated that infiltration of macrophages to the damaged sites is essential for successful appendage regeneration [43,44]. The deletion of the cells by clodronate liposomes results in wound closure but the permanent failure of regeneration [43,44]. In addition, macrophages have been shown to be necessary for the epimorphic regeneration of organs in African spiny mice, suggesting the importance of the macrophages in the modulation of appendage regeneration [45]. Interestingly, the relative abundance appears to be correlated with the mode of wound healing, either scar formation or scar-free regeneration [6]. Excessive influx of macrophages to the wounded tail of lizards prefers the scar formation instead of scar-free regeneration [6]. A smart experiment that has been performed by repetitive amputation or cauterization of the lizard results in the increase of immune cells in the blastemas and a reduction of tail regeneration. This phenomenon has indirectly uncovered the potential relations between overabundant macrophages and the failure of regeneration [46]. In addition, amputated limb of lizards that heals with scar formation maintains a high quantity of leukocytes in comparison with those in scar-free tail regeneration [31,46,47,48]. The influence of excessive macrophages on tail regeneration of lizards may be attributed to stimulating fibroblast activation and rapid production of collagen fibrils underneath the wound epidermis [47]. Notably, two subtypes of macrophages are dynamically present at injured tissues, the proinflammatory macrophages [M1] and anti-inflammatory M2 subtype [49]. The M2-like macrophages together with Tregs lymphocytes are found to distribute among mesenchymal and epidermal cells of the regenerative tail blastema [50,51,52].

## 4. Self-Blocking of the Proinflammatory Signal Pathway in Leukocytes

Phagocytes are evolutionarily retained from invertebrates to mammals, having the ability to wall-off invading microbial agents [53]. The macrophage seems to be more ancient than other granulocytes [53]. Accordingly, the pattern-recognition receptors (PRRs) associated with the innateimmunity are also functionally conserved and expressed in the granulocytes of diverse animals in the phylogeny [54]. PRRs recognize microbial components, known as pathogen-associated molecular patterns (PAMPs), to activate specific signaling pathways and lead to distinct anti-pathogen responses [55]. Furthermore, they can interact with the damage-associated molecular patterns (DAMPs) to initiate cellular inflammatory responses [56]. PRRs include Toll-like receptors (TLRs), Nod-like receptor, and C-type lectin receptor families, among which TLRs have the maximum number of transcripts in lizard splenic transcriptome [57]. This implies that TLRs in immune cells of lizards play major roles in sensing and responding to invading pathogens or damaged tissues.

The tail amputation of a lizard triggers limited inflammatory responses in comparison with those of limb injury [58]. For instance, several DAMPs, including proinflammatory (high mobility group box 1) HMGB1 and MIF, are observed to be upregulated and macrophages are observed to be accumulated at injury sites. However, they do not elicit the inflammatory responses [43,59,60]. Two paralogs of HMGB1, HMGB1a and HMGB1b have been identified in the gecko, and HMGB1b is a response to the tail injury. Both subtypes are not able to interact with TLR2/TLR4 receptors to activate the inflammatory responses of macrophages. Instead, they can bind with the receptor for advanced glycation end products (RAGE)receptor to promote migration of tissue cells (oligodendrocytes) involved in tail regeneration [60]. The biological importance of oligodendrocytes in the tail regeneration of lizards refers to Tokuyama et al. [61]. The distinct biological property of evolved HMGB1 protein in lizards has circumvented its proinflammatory action following tail amputation.

Injury to the spinal cord always initiates excessive inflammatory responses in mammals. However, the severed spinal cord of gecko following tail amputation suffers limited inflammation as determined by an antibody array of various cytokines [59]. The investigation of the underlying mechanism has shown that upregulated expression of suppressor of cytokine signaling-3 (SOCS3) in the macrophages/microglia negatively regulates the proinflammatory signals, which are evoked by granulocyte/macrophage colony-stimulating factor (GM-CSF) and interferon-gamma (IFN-γ). Inflammatory cytokine-induced expression of SOCS3 is able to bind with janus kinase 1/2 (JAK1/2), which in turn suppresses the GM-CSF/IFN-γ-driven inflammatory responses, and thereby serves as inflammatory brake to control macrophage-derived inflammation [59]. 

Astrocytes are regarded as another inflammation-related cell type in the central nervous system (CNS). In mammals, DAMPs, including MIF, can robustly activate the inflammatory responses of astrocytes [62,63]. However, MIF from gecko is not able to facilitate the inflammatory response of gecko astrocytes. Instead, they activate the inflammatory response of rat astrocytes [64]. By transcriptome analysis of MIF-stimulated astrocytes from gecko and rats, it has been revealed that vav guanine nucleotide exchange factor 1 (VAV1) plays a key role in inhibiting MIF-mediated inflammation of gecko astrocytes [64]. Taken together, several negative regulators of inflammation are responsive to the tail injury of lizards by contributing to the control of proinflammatory activation.

It is interesting to note that proinflammatory mediators cyclooxygenases 2 (COX2) and prostaglandin E2 (PGE2) are upregulated at the injured site following gecko tail and limb amputation. However, the different outcomes are observed with non-inflammatory activation in the damaged tail but proinflammatory activation in the damaged limb [65]. During the early healing phase of the tail, EP4 receptor of PGE2 is simultaneously increased, and PGE2-EP4 interaction is correlated to early resolution of inflammation by increasing the anti-inflammatory mediator interleukin-10 (IL-10) and decreasing proinflammatory mediators inducible nitric oxide synthase (iNOS), tumour necrosis factor-alpha (TNF-α), IL-6, IL-17 and IL-22. In contrast, the expression of EP2, rather than EP4, is upregulated in the limb. The binding of PGE2 with EP2 leads to the increase of proinflammatory mediators and the reduction of anti-inflammatory mediators [65]. Such a strategy of the lizard for preventing the activation of inflammation from a tail injury is likely ancient and very efficient.

## 5. Crosstalk between Inflammation and Blastemal Cells or Immature Somatic Cells

The origin of blastemal cells in epimorphic regeneration of vertebrates results from dedifferentiated somatic cells and/or resident stem cells. Following rapid proliferation, these mesenchymal blastemal cells with high heterogeneity differentiate to multiple tissues in a lineage-restricted manner, though the possible existence of transdifferentiation (salamander) [66], as is shown in the formation of cartilage and muscle [67]. To date, the investigation regarding the lineage differentiation of blastemal cells in the lizard is elusive.

Accumulating evidence has shown that acute inflammation induced by appendage injury is responsible for initiating regeneration of organs or tissues [68]. For example, transient upregulation of IL-1β is necessary for normal fin regeneration of zebrafish by inducing the expression of regeneration-related genes, but a prolonged high level of IL-1β is found to be detrimental for regeneration [69]. The complement of this is a primordial sentinel of the innate immune response that engages in multiple inflammatory cascades, and crosstalk is complementary with cytokine and growth factor signaling pathways that drive tissue regeneration [70]. 

Hyaluronic acid (HA), an important immunosuppressive, is largely produced in blastema of lizards, amphibians and fish, where it has the role of inhibiting immune response [71]. Although inflammation seems indispensable for inducing appendage regeneration of vertebrates, prolonged activation of inflammation will inevitably exert negative effects on the regenerative process. Such consequence is tightly associated with its influence on the blastemal cells. On the one hand, transient inflammation promotes the proliferation of precursor/stem cells [68,72]. On the other hand, it also contributes to the aging of these cells, such as proinflammatory cytokines IL-1β, TNF-α and IL-6 in mediating aging of hematopoietic stem cells (HSCs) if without resolution [73]. In addition, different types of stem cells produce distinct characterizations and responses to inflammatory stimulation [74].

In spite of the possible effects of inflammation on the blastemal cells, both transcriptional and translational determination of a lizard’s damaged tail displays that the tail regeneration is under control of limited inflammatory responses [58,59,75,76]. How the regenerating tail of a lizard resolves the inflammation is still unclear. By referring to the results from transplantation of stem cells in various diseases, it can be concluded that stem cells themselves play key roles in mediating anti-inflammatory responses. For example, bone marrow-derived mesenchymal stem cells (BMSCs) alleviated inflammatory responses of allergic airway disease through secretion of Gal-1 [77]. At the site of tissue injury, MSCs can repair tissue damage by producing growth factors and immunosuppressive molecules to inhibit inflammatory cytokine storms [78]. They also attenuate LPS-induced inflammatory responses by inhibiting the NFκB signaling pathway [79]. Taken together, stem cells exert their beneficial effects not only by cell replacement but also by immunomodulation and trophic support following tissue damage [80]. But the direct evidence of blastemal cells playing anti-inflammatory roles during the lizard tail regeneration is lacking.

Comparative analysis of transcriptome profiles of gecko and rat astrocytes revealed that adult gecko astrocytes display a profile of neural stem cells (NSCs) markers similar to those of embryonic rat astrocytes, suggesting an immature cell trait [64]. Interestingly, gecko astrocytes are insensitive to proinflammatory stimulation of MIF and LPS. Vav1, which is abundantly expressed in the gecko astrocytes, contributes to suppressing the inflammatory activation of gecko astrocytes, and its deficiency is able to restore sensitivity to inflammatory stimuli [64]. These results indicate that the immature somatic cells of the lizard are resistant to the various inflammatory stimuli following tissue damage.

## 6. Conclusions

The natural evolution of lizards has endowed them with the powerful capacity of regenerating damaged tails by avoiding infection and inflammation. The lizards can heal the wound rapidly through minimal bleeding and efficient hemostasis to reduce the risks of infection and subsequent inflammation. Furthermore, they have established a protective barrier using the antimicrobial peptides. To block DAMPs-mediated inflammatory activation on the leukocytes at injury sites, lizards evolved the distinct functions of DAMPs that are unable to bind with the PRRs or upregulate the negative regulators of innate immunity in leukocytes. The blastemal cells and immature somatic cells of lizards are resistant to the various inflammatory stimuli following tail amputation. Therefore, tail-regenerative lizards are observed to apply a comprehensive and efficient strategy for fighting against excessive inflammation to promote regeneration following injury.

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
