# Peer review of "Self-Control of Inflammation during Tail Regeneration of Lizards"

_jdb, 2021, doi:10.3390/jdb9040048_

Round 1

Reviewer 1 Report

The ms revised is of significant interest for multi-field study of regeneration process. It contains new important data which analyzed for the understanding of this phenomenon. I offer to publish this ms in the present form.

Author Response

For reviewer1.

The ms revised is of significant interest for multi-field study of regeneration process. It contains new important data which analyzed for the understanding of this phenomenon. I offer to publish this ms in the present form.

Thank you very much for your appreciation of our manuscript.

Author Response

For reviewer2.

This is an excellent review. I read it with great pleasure and interest. Maybe I am not so familiar with the species-specificity of inflammation in lizards, but it seems to me that the argument was presented in a clear way with no faults.

Strong points: it has been clearly shown and demonstrated that there is an evolutionary adaptation leading to a control of normal inflammatory processes, induced upon tissue damage.

It is interestingly highlighted how normal DAMP in lizards can have a modified receptor specificity.

which lead to a reduced inflammatory effect and an increased anti-inflammatory action.

Weak point: it would be excellent to create a picture with basic mechanism of tail regeneration an inflammation control, summarized.

Thanks for the good suggestion. We understand that an illustration of the anti-inflammation mechanism of the lizards will be helpful for the reading of the manuscript, but we are not good at drawings. We hope this should be understood. 

Reviewer 3 Report

I congratulate to the authors writing interesting paper on the regeneration process. It is not often that lizard tail regeneration is taken into account when  studying the regeneration.

Lizard tail regeneration model is interesting, effective and less harmful for the animal than other, especially those in mammalian animals. It also provides possibility to examine different tissues from all developmental lines.

It is good to read a paper where important aspect of inflammation influence on regeneration proces is taken together as a review.

Author Response

For reviewer3.

I congratulate to the authors writing interesting paper on the regeneration process. It is not often that lizard tail regeneration is taken into account when studying the regeneration.

Lizard tail regeneration model is interesting, effective and less harmful for the animal than other, especially those in mammalian animals. It also provides possibility to examine different tissues from all developmental lines.

It is good to read a paper where important aspect of inflammation influence on regeneration proces is taken together as a review.

Thank you very much for your appreciation of our manuscript.